# Gut Microbiota, LADA, and Type 1 Diabetes Mellitus: An Evolving Relationship

**DOI:** 10.3390/biomedicines11030707

**Published:** 2023-02-25

**Authors:** Andrea Piccioni, Federico Rosa, Sergio Mannucci, Federica Manca, Giuseppe Merra, Sabrina Chiloiro, Marcello Candelli, Marcello Covino, Antonio Gasbarrini, Francesco Franceschi

**Affiliations:** 1Department of Emergency Medicine, Fondazione Policlinico Universitario A. Gemelli IRCCS, 00168 Rome, Italy; 2Facoltà di Medicina e Chirurgia, Università Cattolica del Sacro Cuore, 00168 Rome, Italy; 3Section of Clinical Nutrition and Nutrigenomic, Department of Biomedicine and Prevention, University of Tor Vergata, 00133 Rome, Italy; 4Dipartimento di Medicina e Chirurgia Traslazionale, Università Cattolica del Sacro Cuore, 00168 Rome, Italy

**Keywords:** gut microbiota, type 1 diabetes mellitus, LADA

## Abstract

There is much evidence confirming the crucial role played by the gut microbiota in modulating the immune system in the onset of autoimmune diseases. In this article, we focus on the relationship between alterations in the microbiome and the onset of diabetes mellitus type 1 and LADA, in light of the latest evidence. We will then look at both how the role of the gut microbiota appears to be increasingly crucial in the pathogenesis of these disorders and how this aspect may be instrumental in the development of new potential therapeutic strategies that modulate the gut microbiota, such as probiotics, prebiotics, and fecal microbiota transplantation.

## 1. The Gut Microbiota and the Intestinal Barrier

The gut microbiota is a complex ecosystem consisting of bacteria, fungi, archaea, and viruses living in symbiosis with the human organism. This collection of bacteria, archaea, and eukaryotes that colonize the digestive system has developed this fascinating symbiotic relationship with its host over thousands of years, characterized by a complex mutually beneficial interaction [1].

The total number of these microorganisms is estimated to be between 10^13^ and 10^14^, a number close to that of all cells in the human body [2]. Moreover, the genetic makeup of all these microorganisms, called the microbiome, turns out to be even greater than that of humans [3].

Only in recent years, supported by some very important discoveries, and with the fundamental contribution of metagenomics and 16S ribosomal RNA gene sequencing, has the composition and numerous functions of the gut microbiota been better studied and understood [4].

The gut microbiota is physiologically composed of Firmicutes, Bacteroides, Proteobacteria, Actinobacteria, Euryarchaeota, and Verrucomicrobia [5] (Figure 1).

The most represented bacterial phyla are Bacteriodetes and Firmicutes, which make up more than 90 percent of the gut microbiota [6].

The intestinal barrier constitutes a protective defense against pathogens, and toxic and dietary compounds [7]. It is formed by an external epithelial layer and an inner endothelial layer which form the gut epithelial and vascular barrier, respectively [8]. The microbiota reside in the gut lumen at an adequate distance from gut the mucosa [9].

In some contexts, this barrier function is lost in what is called Leaky Gut Syndrome, a condition that has been increasingly discussed lately, which some authors say contributes to the onset of major autoimmune diseases such as celiac disease, rheumatoid arthritis, multiple sclerosis, and type 1 diabetes mellitus [10].

The development of the gut microbiota is one of the most interesting and current fields of research. It is certainly affected by environmental influences, the best known of which is the diet [11].

As a confirmation of the symbiotic role between the gut microbiota and the host, early microbiota colonization right after birth deeply influences immunological, metabolic, and allergic diseases [12].

The gut microbiota physiologically plays a symbiotic role with its host, in which it is involved in numerous functions. For example, it plays an immune role in the defense against pathogens. It also plays a key role in nutrition, being involved in the extraction of short-chain fatty acids (SCFAs) and amino acids from foods [13].

Unhealthy diets and obesity are widely known as risk factors for the development of type 2 diabetes [14]. Alterations in the gut microbiota and their repercussions at the metabolic level play a key role in the development of type 2 diabetes mellitus [15,16]. Changes in microbiota composition and gut dysbiosis increase gut inflammation and directly induce endotoxemia, which is recognized as a conditioning promoter of infectious, chronic, and metabolic diseases and has also gained importance in cancer pathogenesis [17]. Recent studies have highlighted the link between type 2 diabetes mellitus and metabolic endotoxemia. In type 2 diabetes mellitus, metabolic endotoxins are associated with a worsening of glycemic levels [18].

Dysbiosis in the microbiota is associated, with varying degrees of evidence, with a large number of diseases, affecting the gastrointestinal system and others. Those affecting the digestive system include inflammatory bowel disease and irritable bowel syndrome; other disorders include metabolic diseases such as obesity and diabetes, allergic diseases, and neurodevelopmental diseases [19].

## 2. Overview of Diabetes Mellitus

Diabetes is a disease of the endocrine system characterized by high blood glucose levels, which can be recognized by several underlying conditions. The predominant form appears to be type 2 diabetes, which accounts for 90–95% of cases [20].

Diabetes is one of the most prevalent diseases globally: in 2014, 9.3 percent of Americans were reported to have diabetes (29.1 million people) [21].

The incidence of this disease is steadily increasing, and it is estimated to affect 693 million adults globally by 2045, a 50 percent increase from 2017 [22].

The main cause of morbidity and mortality of this disease is due to its micro- and macrovascular complications, such as cardiovascular disease, nephropathy, retinopathy, and diabetes-related neuropathy [23].

Type 1 diabetes mellitus (T1DM), is mainly characterized by insulin deficiency, while type 2 diabetes mellitus (T2DM) is characterized by insulin resistance with relative insulin deficiency. Type 1 diabetes mellitus, also known as juvenile diabetes, remains the most common form of diabetes in childhood, although the rate of type 2 diabetes mellitus is steadily increasing [24].

### 2.1. Type 1 Diabetes Mellitus

In most patients with type 1 diabetes mellitus, destruction of pancreatic beta cells that secrete insulin occurs through an autoimmune mechanism. This mechanism, mediated by autoantibodies, is sometimes referred to as type 1A diabetes [25].

In clinical practice, the assay of autoantibodies against GAD65 (glutamic acid decarboxylase 65), IA2 (40 K fragment of tyrosine phosphatase), insulin, and ZnT8 (zinc transporter 8) is indicated [26].

Type 1B diabetes refers to a minority of patients who, although they have the clinical features of type 1 diabetes mellitus, have no detectable autoantibodies, no evidence of autoimmune beta-cell destruction, and in whom no other causes have been identified. Another characteristic feature of type 1 diabetes mellitus is inappropriately low levels of insulin and C-peptide relative to plasma glucose concentrations, while high levels of fasting insulin and C-peptide point toward a diagnosis of type 2 diabetes mellitus [27].

In the pathogenesis of type 1 diabetes mellitus, many researchers have speculated that a key role may be played by viruses according to several mechanisms. According to some, viruses act by destroying beta cells, while for others they activate an autoimmune-type response directed against these insulin-secreting cells [28].

The viruses most often called into question are enteroviruses (such as Coxsackie B1 and B4), CMV, rubella, and mumps [29].

### 2.2. Latent Autoimmune Diabetes in Adults (LADA)

There is a minority of patients who develop type 1 diabetes mellitus in adulthood, often initially diagnosed as type 2 diabetes mellitus. These patients of older age may have circulating autoantibodies directed against pancreatic beta-cell antigens (ICA or GAD65) [30].

Latent autoimmune diabetes in adults (LADA) is diagnosed in these patients. From the European multicenter study “Action LADA,” it was found that nearly 10 percent of patients with diabetes mellitus arising in adulthood had autoantibodies directed against pancreatic islets [31]. These adult patients may not initially require insulin treatment, but it becomes necessary months if not years after diagnosis [31].

## 3. Diabetes Mellitus and Gut Microbiota

### 3.1. Early Evidence

Researchers have long hypothesized that patients with type 1 diabetes mellitus might have a different composition of the gut microbiota than that of healthy people [32].

This hypothesis is related to the finding of an increased incidence of type 1 diabetes mellitus diagnoses in developed countries, so it was thought that environmental factors might be involved in the pathogenesis [33].

The hygiene hypothesis, which correlates improved hygiene conditions with increased incidences of autoimmune diseases, has been known for some time.

In experiments conducted on laboratory mice, an increased incidence of diabetes was found in those raised in a germ-free environment [34].

It is known that there is a profound link between gut microbiota and the immune system, the dysregulation of which underlies many important diseases.

Dysbiosis is hypothesized to be the cause of dysregulation of the development of the inflammatory response underlying the onset of inflammatory bowel disease [35].

Dysregulation of the immune system is also implicated in the onset of other major autoimmune-type diseases, such as rheumatoid arthritis, psoriasis, multiple sclerosis, and type 1 diabetes mellitus [36].

This evidence is supported by another paper from 2018 [37] that started from the premise that the incidence of diabetes mellitus type 1 has been increasing in recent years, as have those of other immune-related diseases such as celiac disease and allergic diseases.

To explain this phenomenon, the authors hypothesize that underlying it is a decrease in the presence of certain microorganisms, such as *Bifidobacterium infantis*, which plays a critically important role in breast milk metabolism, due to several reasons such as an increase in cesarean deliveries.

This observation is reinforced by another important study conducted in Denmark [38], in which an increased incidence rate of type 1 diabetes mellitus in children who used broad-spectrum antibiotics in the first two years of life was found to be correlated with the mode of delivery. Additionally, it was found that infants delivered by cesarean section and treated with antibiotics had an increased risk of developing type 1 diabetes mellitus. These data reveal the key role played by the gut microbiota from the earliest moments of life. We have seen how its alteration could have such repercussions as to lead to the development of such an important disease as diabetes mellitus. Alterations, as already discussed, can of course come to develop after broad-spectrum antibiotic therapy, just as is the case with the onset of clostridium difficile infection or even secondary to the mode of delivery, again underscoring the key role played by the microbiota from the earliest moments of life.

### 3.2. The Link between Onset of Diabetes Mellitus Type 1 and Gut Microbiota

Some important findings help us to better understand the link between type 1 diabetes mellitus and the gut microbiota.

In autoantibody-positive patients, an increase in the proportion of bacteria belonging to the phylum *Bacteroidetes* was found, while in healthy subjects, that of *Firmicutes* was found to be increased [39].

This interesting article summarizes the main evidence regarding the different compositions of the gut microbiota between patients with type 1 diabetes mellitus and healthy patients [40] (Figure 2).

Furthermore, in patients with type 1 diabetes mellitus, a decrease in the microbiota producing SCFAs (short-chain fatty acids) has been found [41], and for this reason some authors, as we shall see, hypothesize that its replenishment could slow down beta-cell destruction.

This hypothesis is also supported by the finding in murine experiments in which fecal microbiota transplantation (FMT) decreased the incidence of type 1 diabetes mellitus only in those transplanted with selected feces with an abundance of *Akkermansia muciniphila*, a strain that produces SCFAs [42].

We have previously mentioned the link between endotoxemia and the development of metabolic type diseases such as type 2 diabetes mellitus. Gut epithelial and endothelial cell–cell junctions are composed of transmembrane proteins called tight junctions (TJs) [43]. Inflammatory insults to the intestinal barrier are able to dismantle these connection proteins, increasing intestinal permeability, a condition also known as “Leaky Gut Syndrome”. Obesity and unhealthy diets are promoters of increased gut leakiness, and the resulting low-grade inflammation is called metabolic endotoxemia, a condition which is associated with diabetes mellitus onset and complications [44,45].

This evidence comes as a result of Cani’s seminal work in 2007 in which he highlighted how poor nutrition increased metabolic endotoxemia with harmful consequences for the body [46]. This fundamental role from a leaky gut has also been proposed for type 1 diabetes mellitus in which increased intestinal permeability has been found not only in the disease state but also in its preclinical stages [47].

### 3.3. The Role of Short-Chain Fatty Acids and HLA in Diabetes Pathogenesis

There is a complex interplay linking diabetes pathogenesis, and the microbiota and their metabolic products. With regards to these products, short-chain fatty acids (SCFAs) play a pivotal role. Short-chain fatty acids, including butyrate, are products of fermentation of food residues. Butyrate appears to play a key role in metabolism, modulating satiety, blood sugar, and cholesterolemia [48].

The report that short-chain fatty acids play a protective role against diet-induced obesity and insulin resistance has been described in the literature [49].

In addition, oral administration of sodium butyrate in mice significantly increased plasma insulin [50].

According to some theories, diabetes mellitus type 1 may develop due to an alteration of the intestinal epithelial barrier, in the modulation of which short-chain fatty acids play a key role. 

In experiments in non-obese diabetic mouse models, there was an enhancement of beta cells in those that received butyrate and acetate [51,52].

In light of this evidence, the possible role of prebiotics and fecal microbiota transplantation in preventing diabetes complications has been suggested [53,54].

The microbiota may also, through the production of short-chain fatty acids, have an impact on diabetes complications such as atherosclerosis and peripheral artery disease, as SCFAs are believed to reduce inflammation and thus have a negative effect on the formation of atherosclerosis [55].

Some very other interesting piece of evidence comes from the TEDDY (the Environmental Determinants of Diabetes in the Young). The TEDDY study, among its many areas of research, was also interested in the relationship between the onset of juvenile diabetes and the gut microbiota. This study revealed the protective role of short-chain fatty acids against diabetes mellitus type 1 [56].

Type 1 diabetes mellitus falls among the autoimmune diseases, and similar to other autoimmune disorders, there is a close association with genetic factors.

Many researchers have wondered whether there might be a link between HLA (human leukocyte antigen) genotypes and the development of the gut microbiota.

A landmark study, conducted in Sweden over two years, involved infants whose fecal samples were collected at one year of age, the time when the autoimmunity that characterizes type 1 diabetes mellitus typically develops.

Fecal composition was correlated with HLA genotype, and it was found that genotype may have an association with the gut microbiota of infants, resulting in the first study in which HLA genotypes were found to correlate with changes in the gut microbiota in human patients [57].

### 3.4. The Use of Probiotics and Fecal Microbiota Transplantation as Treatment for Type 1 Diabetes Mellitus

Probiotics are defined as microorganisms that are beneficial to the body, while prebiotics are non-digestible foods that stimulate the growth of certain bacterial groups in the intestinal tract [58]. Attempts have been made in recent years to modulate the composition of the microbiota through the use of these new therapeutic weapons for the treatment of many diseases, among which type 1 diabetes mellitus could not be left out.

Some important evidence on the use of probiotics comes from a recent study conducted in India, in which probiotic-treated children saw improved glycemic control indicated by a reduction in HbA1c and insulin treatment requirement [59].

This represents an important starting point, although there is still a long way to go in this regard, and many studies are still needed.

With the use of fecal microbiota transplantation (FMT), as opposed to probiotics, one goes about manipulating the composition of the gut microbiota in a significantly more invasive way.

A randomized controlled trial conducted in 2020 by de Groot et al. [41] demonstrated that fecal microbiota transplantation can prolong beta-cell function in newly diagnosed patients with type 1 diabetes mellitus.

Some other important evidence comes to us from a recent study in which two patients with type 1 diabetes mellitus underwent fecal microbiota transplantation [60].

The results were surprising; one of the two no longer required hypoglycemic treatment during the months of follow-up in the study, while the other discontinued insulin treatment, and only took oral hypoglycemic drugs. These patients underwent multiple sessions of fecal microbiota transplantation, indicating a kind of “reinforcing effect.”

On the strength of this evidence and the multiple insights gained over time, fecal microbiota transplantation is being studied as a therapeutic strategy for diabetes mellitus as well. Just to tie in with our statements, we cite the very recent case of a 24-year-old young man with type 1 diabetes mellitus and malnutrition successfully treated with fecal microbiota transplantation [61].

All of this evidence gives hope that better glycemic control can be achieved in the future through these new therapeutic strategies.

### 3.5. Latent Autoimmune Diabetes in Adults (LADA)

It is interesting to see how quickly certain aspects in medicine change when dealing with a topic as interesting and rich in news as the gut microbiota. In a recent article, the established relationship between alterations in the gut microbiota and the onset of type 1 diabetes mellitus was reiterated, while a clear relationship between the microbiome and the pathogenesis of LADA has not yet been shown [62].

All this changed within a few months. The link between autoimmunity and the microbiota has also been explored in the context of latent autoimmune diabetes in adults (LADA).

A study was conducted in 2021 in four groups of people, one with patients with type 1 diabetes mellitus, a second with type 2 diabetes mellitus, another with LADA, and a control group with healthy patients, who were dosed with serum and fecal metabolites [63].

It was found that in the microbiota of LADA patients, there is a decrease in *Faecalibacterium* spp., *Roseburia* spp., and *Blautia* spp.

These are short-chain fatty acid-producing bacteria which, as already discussed in this paper, seem to be involved in diabetes pathogenesis. In conclusion, LADA patients have a change in gut microbiota which closely resembles that of type 1 diabetes mellitus patients and has significant variations compared to the microbiota of healthy people. The role of immunity resulting from the proper development and functioning of the gut microbiota and the onset of LADA has therefore been examined again in some recent work [64], and the role of short-chain fatty acids produced by symbiotic microorganisms, with their regulatory function towards cytokines and T cells, has once again become increasingly crucial.

Finally, with the intention of emphasizing once again how important this promising new line of research is, we cannot fail to mention this very important trial being conducted in China [65] in which the relationship between LADA, berberine (BBR), and the prebiotic inulin will be evaluated. If improved glycemic control is demonstrated in patients treated with this prebiotic, it would be further confirmation of the enormous therapeutic potential from the careful study of the microbiota.

## 4. Discussion

As we have seen, step by step, we are getting closer to a better comprehension of the influence that the gut microbiota has on many physiological and pathological processes and its relationship with autoimmune diseases. It is now clear how the gut microbiota is a fundamental constituent of the human organism. This superorganism, whose development is affected by countless factors, is increasingly implicated in the pathogenesis of various diseases [66].

The gut microbiota begins to develop from the earliest months of life, as does our immune system, so the hypothesis of an existing interplay between the two of them is fairly reasonable [67].

We have seen how diabetes mellitus continues to be one of the big issues in public health, both because of its high prevalence and important impact on morbidity and mortality. In recent years, thanks to important insights from researchers and the parallel advancement of technology (just think of the introduction and spread of gene sequencing), we have been increasingly able to delve into the relationship that links these two major chapters of modern medicine.

Interest in the relationship between type 1 diabetes and gut microbiota has grown tremendously in recent years [68].

We are confident that in the coming years there will be decisive steps forward in understanding the role that the microbiome plays in the pathogenesis of this important disease. The pathogenesis of diabetes mellitus is a multifactorial etiology, in which genetic and environmental mechanisms all have their influence. Therefore, when conducting research and attempting to recreate the right environmental substrate, such as by modifying the gut microbiota, if genetic predisposing factors are not also taken into account, many of the results will be inconclusive. This is the real challenge of modern medicine, with an increasingly personalized approach to the individual. Additionally, this is precisely why it is very difficult to conduct research that manages to take into account all the variables involved in the onset and progression of this disease.

Speaking of the gut microbiota, the latest research is focusing on the role of some of its less numerous but no less important components, namely viruses and fungi. One of the latest frontiers is the mycobiome, which includes fungi and yeasts, and its potential connection with type 1 diabetes mellitus [69]. Viroma refers to the viral fraction of the gut microbiota, and researchers are trying to investigate its potential role in autoimmune diseases and also in diabetes mellitus [70].

In this case, however, the connection between these two topics seems to be less distant, having already found a link between viral infections at a young age and the onset of type 1 diabetes mellitus, although more studies are clearly needed on this issue. Deepening the understanding of the gut microbiota will allow new therapeutic approaches to be attempted with the aim of improving patient management of the disease.

Diabetes mellitus includes fearsome microvascular and macrovascular complications [71].

As we shall see, one of the therapeutic goals to be achieved by modulation of the gut microbiota is precisely that of better glycemic control, thereby reducing glucotoxicity and the incidence of the various complications that accompany this terrible disease. A relationship has been shown between alterations in the microbiota both in the classic form of diabetes mellitus, type 1, also known as juvenile diabetes, and in LADA, whose defining characteristic is precisely that it arises in adulthood. This speaks volumes about how complex the world underlying immune activation is in these diseases. In both cases, we have underlying insulin deficiency, that is rapid and dramatic in type 1 diabetes mellitus, or late and slower-progressing in LADA. One of the new challenges in medicine would be precisely to understand why such different expressions of insulin deficiency occur in these two disorders that have similar pathogenetic mechanisms. For now, we have mostly pre-clinical evidence and case reports, but considering the solid rationale, we believe future studies will advance our knowledge. There are multiple strategies with which to interact with the gut microbiota for therapeutic purposes, e.g., the use of probiotics and prebiotics, and fecal microbiota transplantation (FMT), which is certainly the most invasive and but also the one with most potential. Many researchers have taken an interest in the use of prebiotics and probiotics as a supportive strategy for the management of type 1 diabetes mellitus [72], with encouraging results being found, charting a course that needs to be more fully investigated in the future.

The use of fecal microbiota transplantation has now become a viable therapeutic option for clostridium difficile infection (CDI), a dreaded condition that is, at its root, a dysbiosis of the gut microbiota [73,74].

In this regard, a new and interesting systematic review conducted on this topic (FMT and T1DM) [75] also cites some interesting papers that go into the direction we have been discussing so far. Although the evidence is still somewhat limited at present, we believe future studies will shed light on this promising field.

## 5. Conclusions

We are still scratching the surface of the underworld which links our immune system and our microbiota. Every day there are new discoveries in the vast world surrounding the gut microbiota, and this is particularly evident in its relationship to the onset of type 1 diabetes mellitus and latent autoimmune diabetes in adults (LADA) as they are related to a dysfunction in the immune system.

Our immune system is profoundly influenced by external factors, and the importance of the microbiome as one such factor is being increasingly recognized. These aspects are particularly fascinating for several reasons.

First, it could help identify those at risk of developing diabetes at an early age and create a screening program.

Then, clearly, it could suggest an integrated approach to treatment. Interventions on the microbiome could alter the course and natural history of diabetes in young patients, delaying insulin therapy and helping preserve pancreatic beta cells.

If risk factors for the occurrence of these diseases could be identified, interventions could be taken to prevent their occurrence. In addition, we have seen how recent findings on the link between the gut microbiota and these diseases are being exploited to develop new therapeutic strategies, such as the use of prebiotics and fecal microbiota transplantation.

The era of precision medicine is approaching, and the microbiome appears to be destined to play a key role.

## Figures and Tables

**Figure 1 biomedicines-11-00707-f001:**
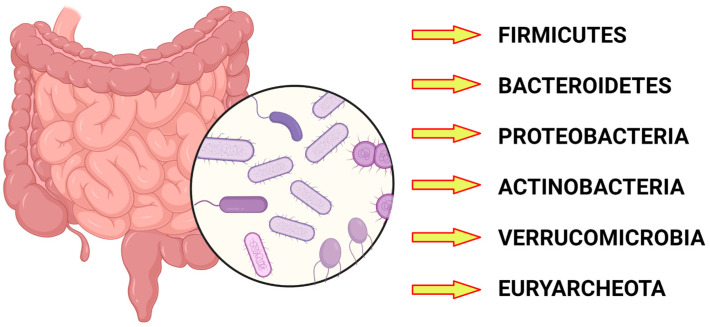
The main phyla constituting the gut microbiota. Created with BioRender.com.

**Figure 2 biomedicines-11-00707-f002:**
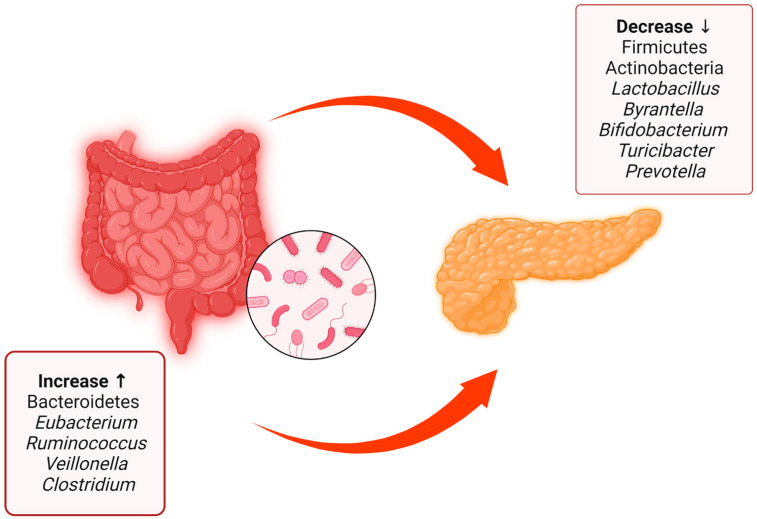
Summary of the main differences between the gut microbiota of patients with type 1 diabetes mellitus and healthy patients, with data obtained from both human patients and rats. There is a decrease in some bacterial species and phyla such as Firmicutes, Actinobacteria, *Lactobacillus*, *Byrantella*, *Bifidobacterium*, *Turicibacter*, and *Prevotella* accompanied by an increase in Bacteroidetes, *Eubacterium*, *Ruminococcus*, *Veillonella*, and *Clostridium*. Created with BioRender.com.

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
