# Peer review of "Gut Microbiota, LADA, and Type 1 Diabetes Mellitus: An Evolving Relationship"

_biomedicines, 2023, doi:10.3390/biomedicines11030707_

Round 1
Reviewer 1 Report
- Please write the name of figures below the figure
- Discussion should be rewritten to sounds good
- Conclusion needs to be more comprehensive with more details
Author Response
- Please write the name of figures below the figure
We did it
- Discussion should be rewritten to sounds good
We did it
- Conclusion needs to be more comprehensive with more details
We did it
Thank you very much for your advice, we have improved our paper and are available for any new suggestions.
Reviewer 2 Report
This review is very interesting, but in this form confusing to me.
First I miss a glossary. In line 137 FMT is used and not explained that happens in line 282. It looks as if different authors has written different parts.
And especial the part on Fecal microbiota transplantation as treatment for Type 1 diabetes mellitus is confusing to me. At line 162 the sentence starts with “Our primary endpoint etc. Who are the “OUR” or is this sentence “cut and paste” from another place ???
And line 192. T1d is used without explanation. So especially this important part 3.4 . of the paper has to be improved.
What is also confusing is that the number of the reference is after the point, it should be before. Now it gives the impression that the reference belongs to the sentence coming.
Again the subject is very interesting but the paper must be improved.
Author Response
This review is very interesting, but in this form confusing to me.
First I miss a glossary.
-We added it.
In line 137 FMT is used and not explained that happens in line 282. It looks as if different authors has written different parts.
And especial the part on Fecal microbiota transplantation as treatment for Type 1 diabetes mellitus is confusing to me. At line 162 the sentence starts with “Our primary endpoint etc. Who are the “OUR” or is this sentence “cut and paste” from another place ???
-We correct it
And line 192. T1d is used without explanation. So especially this important part 3.4 . of the paper has to be improved.
What is also confusing is that the number of the reference is after the point, it should be before.
Now it gives the impression that the reference belongs to the sentence coming.
-We correct it
Again the subject is very interesting but the paper must be improved.
-Thank you very much for your advice, we have improved our paper and are available for any new suggestions.
Round 2
Reviewer 2 Report
The article can be accepted, thank you for the corrections.
Author Response
Thank you for your valuable suggestions, best regards